# Gender Differences in Response to COVID-19 Infection and Vaccination

**DOI:** 10.3390/biomedicines11061677

**Published:** 2023-06-09

**Authors:** Kawther Zaher, Fatemah Basingab, Jehan Alrahimi, Kholood Basahel, Alia Aldahlawi

**Affiliations:** 1Immunology Unit, King Fahd Medical Research Center, King Abdulaziz University, Jeddah 21859, Saudi Arabia; fbaseqab@kau.edu.sa (F.B.); jalrehimi@kau.edu.sa (J.A.); kbasahel@kau.edu.sa (K.B.); aaldahlawi@kau.edu.sa (A.A.); 2Department of Medical Laboratory Sciences, Faculty of Applied Medical Sciences, King Abdulaziz University, Jeddah 21859, Saudi Arabia; 3Department of Biological Sciences, Faculty of Sciences, King Abdulaziz University, Jeddah 21859, Saudi Arabia

**Keywords:** coronavirus disease 2019 (COVID-19), severe acute respiratory syndrome, coronavirus 2 (SARS-CoV-2), sex hormones, pregnancy

## Abstract

Since COVID-19 first appeared, a number of follow-up events have taken place. In an effort to find a solution to this catastrophe, a great deal of study and analysis has been conducted. Because of the high morbidity and exceptionally large losses, scientists are being pushed to conduct more research and find vaccination and treatments. The virus has a wide range of effects, one of which is how it affects sexual activity in both men and women. The impact of the cardiovascular system and susceptibility to embolism, lung stress, and infection heightens the probability of hospitalization in the intensive care unit for pregnant women who have contracted COVID-19. There is no evidence of infection being passed from mother to child. In the current review, the role of COVID-19 infection and vaccination on male and female sexual activity, hormones, and the menstrual cycle for females, as well as on male sex hormones and sexual activity during infection and after vaccination, are being investigated. There are no reports of the virus being isolated from the semen of an infected patient or recently recovered patients. A recent investigation on the influence of the virus on gender susceptibility to sexual organs and function has been uncovered throughout this study.

## 1. Introduction

The novel coronavirus (n-CoV-2) was dubbed COVID-19 in December 2019, with another name, SARS-CoV-2. The World Health Organization (WHO) categorizes it as a severe pandemic disease and advises isolation and lockdown to avoid human-to-human transmission [1]. The propagation of the virus and its epidemiology were unanticipated. The sickness first struck China, followed by Italy and other nations in America and Europe [1]. Despite this, death rates vary widely throughout the country based on population health, vulnerability, accuracy of diagnosis, and emergency response time [2]. COVID-19 is associated with fever, a dry cough, tiredness, and diarrhea. A total of 20% of infected individuals need hospitalization, particularly elderly and chronically unwell patients. In addition, glass-patterned pneumonia may develop, resulting in excruciating chest agony [3].

Mortality included 2.8% of female and 4.8% of male COVID-19 patients. Italy has a male-to-female mortality ratio of 1.79, with men accounting for 65.0% of deaths. COVID-19 killed 1.5–2 men per female. The immunological response of males, females, pre- and post-menopausal women, and children may differ. Children, males, and post-menopausal women showed weaker immune responses than women before menopause. Many reasons induce COVID-19 gender inequalities. These variances are influenced by genetics, lifestyle, comorbidities, hormones, the immune system, and age [4]. The present research investigates how COVID-19 infection and immunization affect male and female sexual activity, hormones, and the menstrual cycle in females, as well as male sex hormones and sexual activity during and after vaccination.

## 2. Origen and Structure of SARS-CoV-2

A toddler with a cold was the first discovered with coronavirus in 1965. Under an electron microscope, the spike glycoprotein on the virus’s surface resembled a crown, thus the name corona. Researchers have identified two essential encapsulating proteins. When a spike or S-protein attaches to cell receptors, cells fuse. M (matrix) is a second glycoprotein that aids in the formation and expansion of the viral membrane [5]. The virus is a single-strand RNA with a 3′-polyA tail and 5′-cap. It has a positive sense range between 26 and 32 Kbp (with an average of 29.8 Kbps). All 29-5′-terminal proteins are encoded by 7–14 open reading frames (ORFs). These ORFs include transcription regulatory areas and leader RNA at the conclusion of transcription. The sub-genomic RNAs ORF1a and ORF1b come together to form a replication–transcription complex. This complex is necessary for building a structure composed of nonstructural proteins (nineteen non-structure proteins) [3,6]. The 3′-terminus of the genome codes for the spike, envelope, membrane, and nucleocapsid proteins [7]. The 150-kilodaltons S protein assists the virus in attaching to the host cell membrane, which contains the viral fusion receptor. S protein is the most prevalent immune ectodomain [8]. S1 attaches to the host cell’s receptors, whereas S2 fuses with cellular membranes. These two components are responsible for the infection of coronaviruses. The trimeric S1 lies atop the S2 stalk (Figure 1). S1 components distinguish coronaviruses [9]. The N terminal domain (NTD) and C terminal domain (CTD) are found in S1 (CTD). CTDs include RBMs (receptor-binding motifs) [10,11]. CoVs-receptor-binding domain (RBD) and its receptor target the angiotensin-converting enzyme2 (ACE2) peptide component [12]. CoV genomes can alter RBD [13]. The viral envelope is formed by structural protein M, having a molecular weight of 25–15,000 kilo Daltons. During replication, M and S (NTD) form the endoplasmic reticulum–Golgi intermediate compartment (E–GIC) complex. The substance diffuses into the endoplasmic reticulum (ER) and gastrointestinal (GI) tracts [14,15]. The E protein, a tiny structural protein ranging in size from 8 to 12 kilo Daltons, contributes to virion assembly, budding, and production [16]. N protein forms a compound with viral DNA to help M protein in viral assembly and transcription [17]. There are three traditional portions. The NTD domain, the first domain, is brief and concise. They are connected electrostatically to the 3′ terminal. The second domain is an RNA-binding domain or serine and arginine-rich flexible linker region (LKR). It influences cellular signaling and interferon production [18]. Incorporating spike proteins into the viral envelope assists viral entrance into specific cells (Figure 1). The surface unit S1 attaches to a cellular receptor, while the transmembrane unit S2 aids the fusion of the viral membrane with the cellular membrane. For membrane fusion, host cell proteases cleave the S protein at the S1/S2 and S2 locations. This cleavage induces membrane-fusing S protein activity [19,20,21]. The constitutive secretory route of infected cells or viral entrance into target cells may break the S protein, which is required for viral pathogenicity [19,22].

## 3. Vaccination

Scientists’ innovation has accelerated COVID-19 vaccine manufacturing. Six vaccine candidates were helped by “Operation Warp Speed.” Among them are messenger RNA (mRNA) (PfizerBNT162b2 BioNTech, Pfizer, Inc., Philadelphia, PA, USA), mRNA-1273 (Moderna vaccines, Moderna TX, Inc., Cambridge, MA, USA), viral vector, and recombined (Figure 2). Moderna received the Food and Drug Administration Emergency Use Authorization (FDA EUA) first. EUA allows the FDA to use unapproved drugs in public health emergencies to prevent, diagnose, or treat life-threatening disorders Upon the realization that a significant number of individuals would be administered the COVID-19 vaccine, the FDA implemented stricter protocols for its review process [23]. Only Moderna, Pfizer-BioNTech, J&J/Janssen, and Novavax have US approval. The CDC recommends the J&J/Janssen COVID-19 immunization only in limited instances due to its adverse effects. The UK has approved Pfizer/BioNTech, Moderna, Oxford/AstraZeneca, Janssen, Nuvaxovid, and Valneva vaccines. Janssen, Moderna, Pfizer/BioNTech, AstraZeneca (SK Bio and Serum Institute of India—Covishield), Sinopharm (China National Pharmaceutical Group, Beijing, China), and Sinovac-CoronaVac (Sinovac Biotech Ltd., Beijing, China) production was approved by WHO as emergency COVID-19 vaccines. These immunizations reduced COVID-19 illness, transmission, and death. The first dose protects for three to four weeks following vaccination. Subsequent doses give better, longer-lasting protection. Most people require a booster dosage for vaccination [24]. European Medicines Agency (EMA) has recommended the use of Astra-Zeneca, Novax, Moderna, Janssen, BioNTech and Pfizer [25]. The AstraZeneca vaccine has been approved and administered in European countries, such as Spain. Conversely, certain nations administered the vaccine solely to select demographics, such as healthcare professionals and educators, who were predominantly female. The company exhibited initial reluctance in issuing it to individuals aged 65 and above due to the limited representation of this demographic in its preliminary clinical trial data [26]. The preliminary investigations indicated a safe vaccine with inevitable minor unfavorable consequences, such as discomfort at the injection site, inflammation, muscle pain, joint pain, and headache. The Medicines and Healthcare Regulatory Agency (MHRA) documented thirteen allergic reactions and six cases of Bell’s palsy, a form of facial paralysis, among patients. It is important to note that there is no conclusive evidence linking these adverse effects to the vaccine, and all affected individuals made a full recovery. The majority of the 143 individuals who were reported to have passed away shortly after receiving the vaccination were elderly patients who had pre-existing health conditions. As a result, these fatalities did not indicate a direct causal relationship between the vaccine and the deaths. Nevertheless, the ongoing assessment of the adverse effects of the vaccines is underway. There have been concerns raised regarding potential adverse reactions associated with the AstraZeneca vaccine, including thromboembolic events and rare instances of blood clotting accompanied by thrombocytopenia and bleeding. This may manifest as cerebral venous sinus thrombosis (CVST) or pulmonary embolism [27]. The aforementioned adverse effects led to the decision of the European Medicines Agency (EMA) to withdraw the vaccine from circulation on 15 March 2021. However, it was reinstated after three days, with the EMA asserting that the advantages of the vaccine supersede its potential drawbacks. Furthermore, the European Medicines Agency emphasized the absence of substantiated data regarding issues on particular vaccine lots. In addition, the agency reported that out of 20 million individuals who were administered the vaccine in Europe, a mere seven cases of disseminated intravascular coagulation (DIC) and eighteen cases of cerebral venous sinus thrombosis (CVST) were observed, resulting in nine fatalities [27,28]. Most of these occurrences were observed in female individuals below the age of 55. Consequently, AstraZeneca has been prohibited from administering to individuals below 60 in Germany and several other European nations. The existence of a causal relationship between the vaccine and these issues has not been substantiated; nonetheless, it has been determined that the occurrence warrants additional investigation [29].

### 3.1. mRNA Vaccines

Moderna and Pfizer-BioNTech created lipid nanoparticle mRNA vaccines. These mRNA vaccines encode SARS-CoV-2 spike protein. This protein is linked to the virus ACE2 to start the infection. Lipid nanoparticles help cells enter injection sites. Host cells transcribe mRNA to make spike protein. B and T cell surface spike protein exposure produces an immunological response [30,31]. Both FDA-approved mRNA vaccines protect against severe COVID-19 and have a robust immune response. Pfizer-BioNTech and Moderna vaccines prevented COVID-19 in 16- and 18-year-olds in phase III studies [32]. The site of injection responses (84.1% for Pfizer/BioNTech and 91.6% for Moderna), weariness (62.9% and 68.5%, respectively), headache (55.1% for both vaccinations), chills (31.9%), fever (14%), joint pain (23.6%), and muscle soreness (38.3%) are among the most common side effects. In addition, mRNA immunization may cause rare myocarditis, allergies, and Bell’s palsy [33,34,35]. If COVID-19 immunization recipients suffer severe allergic responses, the CDC recommends monitoring them for 15 or 30 min. Pfizer-BioNTech’s vaccine is permitted for 12–18-year-olds, although Moderna’s is 18. Both vaccinations must be given twice, 21 (Pfizer-BioNTech, Brooklyn, NY, USA) to 28 (Modena, Cambridge, MA, USA) days apart [36].

### 3.2. Viral Vector Vaccines

Oxford and AstraZeneca’s Johnson & Johnson Janssen’s Ad26.CoV2 deliver spike protein RNA to host cells through viral vectors. The harmless, nonreplicating viral vector has been altered. The SARS-spike CoV-2 protein in the vaccination induces an immunological response like mRNA-based immunizations. Janssen’s single-dose vaccination is authorized, unlike Oxford/AstraZeneca’s. The sole US-approved 18-year-old vaccination is Janssen [37]. The Janssen vaccination may cause pain at the injection site (48.6%), headache (38.9%), weariness (38.2%), and muscle pain (33.2%), although they usually go away within two days. These symptoms are decreased with mRNA-based vaccinations. In addition, Janssen immunization may cause spontaneous radiculitis and Guillain–Barré syndrome. Janssen COVID-19 vaccinations have caused rare cases of life-threatening thrombosis with thrombocytopenia syndrome (TTS) [38]. After 7.98 million vaccine doses, 15 females had severe cases of thrombosis. Pregnancy, hormone replacement treatment, and contraceptives increase thrombosis risk [39]. Thus, women under 50 should be advised that the Janssen vaccination increases thrombosis risk [40].

### 3.3. Recombinant Antigen Proteins

Novavax and GSK-Sanofi employ adjuvant-connected protein subunits to boost immune response. Sanofi has developed VidPrevtyn Beta, a monovalent, recombinant-protein COVID-19 vaccine. This vaccine is designed based on the Beta variant spike antigen and incorporates GSK’s pandemic adjuvant. In 2020, The European Commission approved this vaccine as the first next-generation, protein-based adjuvanted COVID-19 vaccine with its booster dose and approved it to be used for the age of 18 and older [41]. These spike protein-containing vaccines induce an immune response. However, clinical trials prevent the widespread use of these vaccines. Regarding COVID-19, NOVAX reduced mild and moderate illnesses by 96.4% and severe disorders by 100% [42].

### 3.4. Inactivated COVID-19 Virus

WIV04, HB02, and CoronaVac (Sinopharm, Beijing, China) boost immune systems using inactivated COVID-19 virus. Sinovac and Sinopharm employ aluminum hydroxide as adjuvant without live viruses that cannot cause disease. Sinopharm and Sinovac showed 83.5% (based on 40,000 participants) and 73% (based on 10,000 participants; protecting people from COVID-19 without prior infection) efficacy in phase III clinical studies. Both immunizations must be repeated as a booster dose. Injection site reactions, chills, headache, muscular soreness, weariness, joint pain, and fever were expected. Muscle and joint pain were prevalent. China, Brazil, UAE, Chile, Mexico, Indonesia, Hungary, and Turkey sell these two vaccines. However, The US has not approved these immunizations [43]. The side effects following the initial vaccine dose of Sinopharm included normal injection site pain, fatigue, and headache, with a higher prevalence observed among participants aged 49 years or younger compared to those over 49 years of age. In both age groups, the most commonly reported side effects following the second vaccine dose were pain at the vaccination site, fatigue, lethargy, headache, and tenderness. The side effects for both doses were higher among participants aged 49 years or younger. In addition, side effects were higher in females than males for both dosages [44].

## 4. The Interaction of COVID-19′s Infection Endocrine Factors Related to Male Gender and Prostate Cancer

How vaccines impact women and men is based on genetics, hormones, and the dose given. COVID-19 responses varied by gender. Males account for 59–68% of cases and have higher mortality rates. Due to a diminished immune system, those over 75 are more prone to the disease and its repercussions [45]. The SARS coronavirus 2 has distinct effects on men and women (SARS-CoV-2). Males had a more significant hospitalization and mortality rate than women, while women had a higher COVID-19 risk over the long term. The primary risk factors for severe disease are gender, age, and cardio-metabolic comorbidities. Males are twice as likely as women to get a SARS-CoV-2 infection that is severe or deadly [46]. Prepubescent children of both sexes are more resistant to COVID-19 than adults [47]. This shows that gender-linked variables, such as sex hormones, may contribute to the development of COVID-19 [48]. Type 2 SARS coronavirus is an enveloped virus with a single RNA strand penetrating host cells through ACE2. TMPRSS2, a host type 2 transmembrane serine protease, facilitates viral entry by priming the viral S glycoprotein. SARS-CoV-2 may not bind to target cells if its expression is suppressed, but tissue (co-) expression of ACE2 and TMPRSS2 near viral entry sites may increase infection. Sexual steroids influence both sexes genetically. ACE2 is a member of a group of genes that may be resistant to X-chromosome inactivation and are expressed greater in male tissues [49]. Lung expression is primarily masculine. Male-predominant ACE2 expression corresponds to males having more significant ACE2 activity than women, partly because of sex steroids. Plasma from males has higher ACE2 than plasma from women, which may imply tissue expression [50]. Sexual steroids modify ACE2 tissue specifically. For example, Estradiol downregulates kidney ACE2, while ovariectomy, which depletes estrogen, increases ACE2 activity and expression in kidney and adipose tissue in mice [51]. In differentiated airway epithelial cells, estrogen might inhibit ACE2, the primary SARS-CoV-2 entrance site [52]. Conversely, testosterone enhances ACE2 expression in human airway smooth muscle (ASM) cells, which is much lower in women than in men [53]. Men showed lower ACE2 levels in old age than women, consistent with the age-dependent decline in circulating steroid hormones [54]. The findings indicate that the two primary sex steroid hormones oppose ACE2 regulation, supporting the idea. Estrogen reduces the primary SARS-CoV-2 receptor in several organs, while testosterone enhances it. Androgen stimulates the expression of TMPRSS2 in human lung epithelial cells, while its absence inhibits this expression [55]. External testosterone elevated TMPRSS2 expression in type 2 pneumocytes from humans [56]. Thus, men have greater testosterone levels, promoting viral replication and COVID-19 severity [57]. In 118 individuals with primary prostate cancer, androgen deprivation decreased the risk of SARS-CoV-2. The confidence interval for the odds ratio for SARS-CoV-2 infection was 1.55 to 10.05, with a confidence level of 95% [58]. The data suggest that androgen control of TMPRSS2 expression increases COVID-19 risk, but larger cohorts are needed to confirm them. Recently, Samuel et al. highlighted 1443 FDA-approved drugs and an in silico screen of more than nine million drug-like compounds to find medicines that reduce ACE2 protein levels in lung organoids and cardiac cells. The best drugs reduced androgen receptor signaling, according to researchers. Inhibitors of 5-alpha reductase inhibited ACE2 and TMPRSS2 in cardiac cells and lung epithelial, lowering the infectiousness of SARS-CoV-2 in lung organoids. Androgen receptor signaling is inhibited by 5-alpha reductases. ACE2 and TMPRSS2 co-expression in SARS-CoV-2 target tissues may elucidate men’s higher COVID-19 rates. However, further research is required to determine if sex hormones in the lung or other SARS-CoV-2 target tissues control the severity or susceptibility of COVID-19 [58]. It is noteworthy that ACE2 is also expressed in the pancreas, and a recent study has demonstrated an elevated incidence of pancreatic damage after SARS-CoV-2 infections. In this regard, it has been observed that male individuals exhibit a higher susceptibility to the onset of diabetes or pancreatic cancer than their female counterparts [59].

COVID-19 has been observed to potentially cause a temporary reduction in male fertility by causing damage to testicular tissue and/or hindering the process of spermatogenesis. Additionally, compared to healthy men of a similar age, a reduction in the ratio of testosterone/LH and FSH/LH has been observed in individuals with COVID-19 [60].

Recent data suggest a potential correlation between prostate cancer (PCa) and COVID-19, wherein androgen-deprivation therapies (ADT) utilized in PCa treatment exhibit a protective effect against COVID-19. Exploring the potential mechanisms contributing to the interplay between COVID-19 and PCa suggests an apparent correlation between the targets of SARS-CoV-2 on host epithelial cells and the genetic anomalies and molecular markers of PCa, such as AR and TMPRSS2. The inquiry is whether prostate cancer (PCa) treatments could potentially serve as therapeutic options for patients with COVID-19 [61].

### 4.1. The Interaction of COVID-19′s Infection, Female Gender and Breast Cancer

While initial reports from China indicated a higher prevalence of male COVID-19 patients, recent research indicates that females may be more susceptible to the virus. According to data collected by the Korean Society of Infectious Diseases, out of 4212 COVID-19 patients, 37.7% were male, and 62.3% were female. These findings differ from the data reported in China, where approximately 51% of COVID-19 patients were identified as male. A study conducted in Qingdao City, China, analyzed 44 COVID-19 patients and found that 66% were female. In addition, a study conducted in the Zhejiang Province of China analyzed the gender distribution of COVID-19 patients across different age groups, including young and elderly patients [62].

On the other hand, it was observed that COVID-19 patients aged over 60 years were mostly female, with a male-to-female ratio of 43% to 57%. The observed variations in gender distribution may be attributed to the higher prevalence of medical comorbidities reported among elderly patients. Specifically, the older age group had a significantly higher proportion of comorbidities than the younger age group (55.15% vs. 21.93%). Finally, a multicenter study conducted in Europe analyzed 417 COVID-19 patients and revealed that a more significant percentage of female patients (63%) were affected by COVID-19 than male patients (37%). The study found that females diagnosed with COVID-19 exhibited a higher likelihood of experiencing olfactory and gustatory dysfunctions than their male counterparts. The biological process responsible for the sensory dysfunction observed in female patients with COVID-19 remains unclear [63].

According to a study conducted by Kyrou I et al., women diagnosed with polycystic ovary syndrome (PCOS) and multiple cardio-metabolic diseases are at a higher risk of developing Type 2 Diabetes Mellitus, Hypertension, Dyslipidemias, Obstructive Sleep Apnea, and Non-alcoholic Fatty Liver Disease. Additionally, a correlation has been observed between severe cases of COVID-19 and specific factors, including hyper-inflammation, low levels of vitamin D, and hyperandrogenism, all of which are directly linked to polycystic ovary syndrome (PCOS). Hence, there is an intersection between specific shared characteristics of PCOS and recognized risk factors for severe COVID-19. This suggests that individuals with polycystic ovary syndrome (PCOS) may be at an elevated risk for experiencing a severe infection from the SARS-CoV-2 virus [64,65].

Based on the available data, the most recent discourse revolves around the hypothesis that elevated estrogen levels may protect against COVID-19 in breast cancer patients. This has raised several inquiries, including whether endocrine therapy may disrupt the astrobleme and heighten patients’ vulnerability to COVID-19 infection [66]. According to recent studies, estrogen may exhibit a protective effect against COVID-19. An increase in estrogen levels leads to upregulation of Estrogen receptor alfa (ER-α) in T lymphocytes. In addition, estrogen has been observed to enhance the secretion of interferon I and III by T lymphocytes. Elevating interferon I and III levels has been observed to mitigate COVID-19 infection [67].

During the COVID-19 infection, the virus inhibits ACE2 and halts the production of angiotensin 1–7. The production of Angiotensin III may elevate with a decrease in the destruction of Angiotensin II. Therefore, there may be an upregulation of aminopeptidase. Elevated levels of aminopeptidase have been observed to potentially contribute to chemotherapy resistance in breast cancer patients undergoing chemotherapy treatment. The administration of Tamoxifen has been observed to result in the downregulation, mutation, or loss of estrogen receptors. With prolonged usage of Tamoxifen, the drug can permanently affect estrogen receptors. Therefore, in cases where estrogen receptors are impaired or suppressed, the binding of estrogen to these receptors may not occur. Tamoxifen functions as a P-glycoprotein inhibitor, regardless of its impact on estrogen receptors. It inhibits T cell functionality and interferon secretion. Therefore, it is hypothesized that Tamoxifen may elevate the risk of COVID-19 infection due to its antiestrogenic and P-glycoprotein inhibitory properties [67].

### 4.2. Interaction between COVID-19 Infection and Teenagers and Young Adults

In the CDC Report 2020, there was an observed increase in the reported weekly incidence of COVID-19 and the percentage of positive test results among children, adolescents, and young adults. This increase was particularly notable during the early summer months, followed by a subsequent decline and a steep rise from October through December. The patterns observed in the incidence rates and proportion of positive test results among preschool-aged children (0–4 years) and school-aged children and adolescents (5–17 years) were similar to those seen in adults during the summer and fall seasons, even during the period when some schools had resumed or were conducting in-person classes. Furthermore, the reported disease incidence exhibited an upward trend with increasing age among children, adolescents, and young adults. Specifically, the incidence and proportion of positive test results were consistently lower among children aged 0–10 years compared to the older age cohorts. The available case data do not suggest a correlation between the rise in incidence or positivity rate among adults and a prior increase in preschool-aged children, school-aged children, and adolescents. On the contrary, the incidence rate among young adults (18–24 years old) was comparatively higher than other age groups during the summer and fall seasons. The peak incidence was observed in mid-July and early September, which preceded the increase in other age groups. This observation suggests that young adults may have a greater role in community transmission than younger children. The results obtained from the national case and laboratory surveillance data support the existing evidence regarding the risk of transmission in school environments. As of December 7, most K-12 school districts in the United States, specifically 62.0%, provided either complete or partial in-person learning, including hybrid models with virtual components. Despite the prevalence of in-person learning, reports of outbreaks within K-12 schools have been limited, according to the Centers for Disease Control and Prevention (CDC). As of the week beginning December 6, the aggregate COVID-19 incidence among the general population in counties where K-12 schools offer in-person education was similar to that in counties offering only virtual/online education. Multiple school districts in the United States that conduct regular surveillance of in-school COVID-19 cases have reported a lower incidence rate among students compared to the surrounding communities. A recent study has also indicated no significant increase in COVID-19 hospitalization rates associated with in-person education. In contrast to the findings related to K-12 schools reopening, prior research indicates a rise in community incidence in counties where higher education institutions resumed in-person instruction. Furthermore, the case surveillance data presented revealed distinctive patterns. The effective prevention of the introduction and transmission of SARS-CoV-2 in schools is contingent upon implementing mitigation strategies within schools and managing transmission within communities. In settings where community incidence is low and effective mitigation strategies are implemented, studies have shown promising results in controlling secondary transmission within childcare centers and schools. These findings are based on preliminary evidence from in-school transmission studies [29]. Educational institutions offer a well-organized setting that can facilitate compliance with essential mitigation protocols aimed at preventing and reducing the transmission of COVID-19. In situations where community transmission rates are elevated, it is reasonable to anticipate the occurrence of COVID-19 cases within schools. As with any communal environment, schools have the potential to facilitate the spread of the virus [68], particularly in instances where preventative measures, such as consistent and appropriate mask usage, are not established or adhered to. A minimum of four limitations constrain this report’s findings. The incidence of COVID-19 among children and adolescents has probably been underestimated due to lower testing volume in these age groups compared to adults. Additionally, positive test results were generally higher among children and adolescents, particularly those aged 11–17 years, than among adults. Testing was often prioritized for individuals with symptoms, while asymptomatic infection was frequently observed in children and adolescents. It should be noted that the data on race/ethnicity, symptom status, underlying conditions, and outcomes are incomplete, and the level of completeness varies by jurisdiction. As a result, any findings related to these variables may be susceptible to reporting biases and should be approached with caution. The analysis presented herein examines case surveillance data for children, adolescents, and young adults. Studies indicate that the risk of COVID-19 introduction and transmission among children may be lower in childcare centers and elementary schools compared to high schools and institutions of higher education. Additionally, there appears to be a lower incidence of COVID-19 among younger children. This may be attributed to the stress of maturity and irregular sex hormones release. Additionally, data has been released that even among teenagers, young males’ risk is higher than females, and for incidence, it is higher in females [69]. Studies have shown that the risk of COVID-19-related mortality is roughly 1.5 times higher in males. Estimated COVID-19 mortality rates for youth appear to be 6–10 times lower in the Netherlands vs. the US. In countries with low numbers of COVID-19 fatalities in children and young adults, vaccinating youth makes a less convincing case [70].

## 5. Effect of Testosterone on Suppressing the Immune Respond

Testosterone has a pleiotropic impact, inhibiting the immune system and reducing inflammation [71,72]. T cells make up 80–90% of blood lymphocytes. T lymphocytes may be alienated into T-helper, T-suppressor, and T-cytotoxic cells [73]. To understand how gender affects the immune response, researchers tracked blood cell counts throughout the reproductive process. The overall number of T cell receptor-expressing lymphocytes and subsets are similar in men and females. Still, males have fewer T lymphocytes [74]. Males have fewer T cells while having the same number of lymphocytes. Men have higher testosterone levels, which promote T lymphocyte apoptosis [75]. Lymphocyte subtypes and numbers do not alter over the menstrual cycle in women. Estrogen and progesterone do not impact lymphocyte numbers [75]. Male studies contradict this. Several studies [76] have shown that testosterone lowers immunity in males. Since testosterone inhibits type 2 and type 17 T-helper cells, antibody responses and B cell proliferation are diminished, lowering adaptive immunity [77]. These parameters were connected to a delayed antibody response in extremely ill COVID-19 patients and lower IgG production in men than women, which may be linked to a worse prognosis [78]. In dendritic cells, which deliver antigens to humoral immune cells, testosterone lowers interleukin cytokines (IL-10, IL-13, and IL-4) in vitro. It also decreases antigen-presenting cell MHC-II receptor expression [79]. In contrast, testosterone increased anti-inflammatory cytokines in vivo tests, contradicting that testosterone suppresses inflammation [77]. In humans and animals, testosterone upregulates IL-10 and downregulates IL-6, IL-1, and TNF-alpha (TNF-). Mohamad et al. reported these results [80]. Testosterone suppresses type 17 T-helper cells and promotes regulatory T-cells, reducing inflammatory immune response [81]. Age and illness are the leading causes of testosterone reduction in men. Many studies show that testosterone levels in males begin to fall between 0.4 and 2% every year after 30 [82]. Hypogonadism—low testosterone—can occur in older adults. Obesity, diabetes, and cardiovascular diseases are major concerns during this period [83]. Several studies have connected these comorbidities to lower testosterone levels [84]. Low testosterone production in older men with comorbidities increases systemic inflammation [80]. So, older adults and their comorbidities may have lesser anti-inflammatory cytokines than healthy young men. IL-6, TNF-, and IL-1 are also greater in older men with reduced testosterone [85]. In Hamburg, Germany, most COVID-19-infected men exhibited lower testosterone levels [86]. According to case studies, the dead population during COVID-19 was mostly males and under 65 years old [87]. These factors suggest that males may produce more pro-inflammatory cytokines, rendering them more vulnerable to COVID-19. The androgen-dependent expression of TMPRSS2 was also discovered recently. SARS-CoV-2 infection requires TMPRSS2 spike protein priming. Human TMPRSS2 gene promoters have a 15-base-pair androgen response element (ARE). TMPRSS2 may need androgen receptor interaction with these response elements. TMPRSS2 has no other human promoter [88]. In lung epithelial cell lines and prostate cancer, testosterone increased TMPRSS2 expression [56]. Men are more sensitive to COVID-19 because testosterone expression increases TMPRSS2 expression [89]. TMPRSS2 helps proteolytically activate other influenza, coronaviruses, and SARS-CoV-2 [90]. Males with higher TMPRSS2 expression are more likely to contract SARS-CoV [91] and MERS-CoV [92]. Therefore, testosterone-mediated TMPRSS2 modulation may explain male predominance in these infections [90]. Nevertheless, COVID-19 patients have not been studied with anti-androgen therapy. The COVID-19 therapy hydroxychloroquine and its analog chloroquine phosphate diminish testosterone release in albino rats [93]. Further study is needed to discover whether testosterone levels affect anti-viral treatment efficacy.

## 6. Effect of Estrogen on Strengthening the Immune Respond

Estrogen improves women’s immunity [94]. Interfering with auto-reactive B cell adverse selection generates T helper cell responses for adaptive immunity [95]. Estrogen receptors (ERs) are present in the gut epithelial cells, lymphoid tissue cells, brain, and immune cells such as macrophages, monocytes, and lymphocytes [96]. There are estrogen receptors in reproductive organs and elsewhere. The most significant estrogen receptor (ER) is Estrogen receptor-alpha (ER-), which is expressed on all immune cells and governs their maturation [97]. ER signaling in innate immune cells causes sex-specific immunological responses [98]. According to several studies [98], Estradiol stimulates type 1 interferon production following attachment to ER. ER- may directly control type 1 interferon production or enhance the expression of genes implicated in pathways for detecting innate stimuli [99]. Estradiol increases type I interferon-inducible innate pathways and gene expression in mouse splenocytes. Before and after interferon stimulation, researchers discovered that interferon regulatory genes were strongly expressed in females [100]. As seen here, ER and interferon-1 modulate innate immunity. Despite the gender discrepancy, females overexpress ER [101]. Estrogen enhances T cell chemokine receptor 5 (CCR5) expression by boosting blood lymphocyte adherence to particular endothelial cells [89,90]. Estrogen enhances the production of IL-4 and the development of type 2 T-helper cells [102]. Estrogen stimulates the differentiation of T-helper type 1 cells in mice while reducing T-helper type 17 cells and IL-17 cytokine levels [103]. These findings imply that estrogen stimulates the growth of B cells and antibody production in women, enhancing their immunity. In addition, several human and rodent studies have demonstrated that estrogen in physiologically higher concentrations inhibits the synthesis of several pro-inflammatory cytokines involving IL-6, IFN-α, and IL-1, and chemokine (C-C motif) ligand 2 (CCL2) from monocytes and macrophages, thus preventing neutrophils and monocytes from migrating to inflamed areas. Very ill COVID-19 patients showed more significant levels of IL-6 than slightly ill individuals [104]. A postmortem inspection of the lungs of a COVID-19 patient indicated an increase in pro-inflammatory cells, namely macrophages and T-helper cells. The SARS Co-V research on mice also revealed a substantial function for sex hormones in the pathogenesis of coronaviruses. Male mice of the same age as infected female mice exhibited higher inflammatory monocyte-macrophages in their lungs. This was plausible since infected female animals produce fewer sex hormones than male mice. The death rate for gonadectomized female mice was 85 percent, compared to 20 percent for female control mice [105]. According to these studies, increased estrogen signaling in females may avoid cytokine storms by reducing pro-inflammatory chemical production and lowering lung monocyte and neutrophil migration. Hence, a well-balanced immune response in women may afford superior protection against SARS-CoV-2 than in males.

## 7. Correlation between COVID-19 and Genes Responsible for Immune Responses on the X Chromosome

A double X chromosome is an additional important component contributing to the hyper-responsiveness of female immunity. In terms of genetics, X chromosomes include several genes that are important for immunity, such as Fork-head box P3 (FOXP3), CD40 ligand (CD40L), Toll-like receptor 8 (TLR 8), and Toll-like receptor 7 (TLR 7) [106]. It is possible to hypothesize, in the context of gender inequality, that any mutations that affect the genes that are accountable for immunity and associated pathways may alter the immunological responses and signaling pathways in both females and males. Nevertheless, again, this is something that has been hypothesized before. A twofold dose of the proteins connected with the X chromosomes is possible in females. An advantage against deleterious X-linked mutations and confers extra diversity in various biological and immune responses is provided by the random silencing of one X-chromosome during X-chromosome inactivation, a recurring event in female embryogenesis [107]. This process occurs during X-chromosome inactivation, a recurring event in female embryogenesis.

On the other hand, it is not hard to see that, on rare occasions, a few genes on X chromosomes may circumvent the silencing process [108]. For example, an escape in the genes connected to immunity or the immune system causes females to have more immune genes than men because of biallelic expression. This phenomenon has two distinct effects on the physiology of females. First, biallelic expression of X-linked genes in immune cells may cause adverse inflammatory or autoimmune responses or help increase immunity [109]. One such protein is called TLR-7, and it is responsible for the identification of pathogens and the activation of innate immunity. It is primarily expressed in monocytes, plasmacytoid dendritic cells (PDC), and B-cells [110]. The X chromosome is responsible for encoding this gene. If TLR-7 can bypass the silencing imposed by the X chromosome, the biallelic expression of TLR-7 causes it to be overexpressed in females relative to men [111,112]. This might be another reason why females have higher immunity than men and could, as a result, give a protective edge against COVID-19 infection or mortality. Nevertheless, these hypotheses need more investigation and testing before reaching a definitive conclusion.

## 8. Menstrual Cycle Changes Caused by the COVID-19 Infection and Vaccination

During the pandemic, various anecdotal information about the possible effects of COVID-19 on the menstrual cycle has appeared. The following are some of the modifications that have been reported: (1) times with lower levels of activity; (2) times of heightened ferocity and activity; (3) irregular periods; and (4) periods were skipped. Changes in menstruation volume were found in 45 of the 177 people investigated for this study (25 percent). Thirty-six out of the mentioned forty-five persons had a noticeably lighter period, whereas nine had a noticeably heavier period [113]. Patients diagnosed with COVID-19 had an increased risk of having menstrual cycles that lasted longer than 37 days on average [114]. According to this study, long cycles were seen in 34% of patients with severe disease and 19% with moderate illness. When the researchers compared the length of a person’s menstrual cycle during COVID-19 to the length of their usual cycle, they found that 28 percent of the total number of participants, or 50 out of 177, exhibited alterations in their menstrual cycle. During their sickness, the majority of them had cycles that were longer than usual, while some had periods that were shorter than typical. The levels of sex hormones such as estrogen, progesterone, and follicle-stimulating hormone were tested in 91 individuals with COVID-19 and 91 without COVID-19. There is no discernible difference between the two groups.

Finally, the researchers concluded that one to two months after having COVID-19, 84 percent of women had returned to their average menstrual volume, and 99 percent had recovered to their standard cycle length [115].

People may have the following adverse reactions after receiving the COVID-19 vaccination: tiredness; aches and pains; soreness; redness or swelling at the injection site; headache; fevers or chills; and nausea. After receiving the second dosage of the vaccines made by Pfizer-BioNTech and Moderna, adverse effects are often more severe. This is entirely natural and a sign that their bodies are working on building up their immunity. COVID-19 immunizations have sometimes been known to cause side effects, including a severe allergic reaction. This often occurs shortly after the vaccination, so they should be monitored for a few days following the event. The Johnson & Johnson immunization has a negligible possibility of producing life-threatening blood clots in its recipients. This can happen weeks after receiving the vaccination, and according to this study, it is more prevalent in women under 50. The Centers for Disease Control and Prevention (CDC) and the Food and Drug Administration (FDA) report that the likelihood of encountering this adverse effect is very low [116]. However, almost immediately after the administration of the COVID-19 immunization, concerns were voiced about the possible connection between the vaccine and alterations in the menstrual cycle. In addition to this, the proliferation of tales on social media caused a good number of women to question whether or not the changes in their menstrual cycle may be connected to the vaccination, and even those suspicions caused some of them to avoid getting the vaccine [117].

In another trial, 3959 people participated in this study (unvaccinated 1556; vaccinated 2403). The Pfizer-BioNTech vaccine was taken by the majority of the cohort (55%), followed by the Modera vaccine (35%), and then the Johnson & Johnson/Janssen vaccine (7%). Overall, the COVID-19 vaccine was accompanied by a change in cycle length of less than one day for both vaccine-dose cycles compared with pre-vaccine cycles (first dose 0.71 day-increase, 98.75% CI 0.47–0.94; second dose 0.91, 98.75% CI 0.63–1.19). On the other hand, unvaccinated individuals did not see a significant change when compared with the first three baseline cycles (cycle four 0.07, 98.75% CI—the difference in change in cycle length between the unvaccinated and vaccinated cohorts was less than one day in adjusted models for both doses (difference in change: first dose 0.64 days, 98.75% CI 0.27–1.01; second dose 0.79 days, 98.75% CI 0.40–1.18)). In addition, the change in cycle length between the unvaccinated and vaccinated cohorts was less than one day. There was no correlation between vaccination and a change in the menstrual cycle duration [117].

In Saudi Arabia, 4170 people were asked for their feedback. Of these, 2601 individuals received one dosage of BNT162b2, of which 456 individuals finished the second dose, and 1569 individuals received a single dose of ChAdOx1. Those who willingly replied to a survey on post-vaccination adverse reactions reported experiencing the adverse effects 85.6% of the time after receiving the BNT162b2 vaccine and 96.05% after receiving the ChAdOx1 vaccine. When compared to ChAdOx1, the adverse effects of BNT162b2 were more severe. A total of 30.13%, 28.62%, 29.73%, and 1.53% of people who received the ChAdOx1 vaccination reported experiencing mild, moderate, severe, or critical adverse effects, respectively. In contrast, most people who received the BNT162b2 vaccination (63.92%) reported experiencing only mild side effects, whereas the percentages of those with moderate, severe, or critical adverse effects were, respectively, 27.67%, 7.68%, and 0.72%. Compared to BNT162b2 vaccination recipients, ChAdOx1 vaccines were shown to have a higher incidence of both systemic and local adverse effects. In the most recent research, palpitations were one of the new adverse responses participants experienced in high frequency. In addition, an abnormal menstrual cycle was recorded in 0.98% (18/1846) of Pfizer-BioNTech vaccinees and 0.68% (7/1028) of ChAdOx1 vaccines. However, deep vein thrombosis was only reported in a single instance of a patient vaccinated with the BNT162b2 vaccine. According to the research findings, both vaccinations were associated with post-vaccinal side effects; however, ChAdOx1 was associated with a greater incidence of post-vaccinal systemic adverse effects than BNT162b2 [118].

## 9. Effect of COVID-19 on Pregnancy

Reports by the CDC in the United States detail that the risk of COVID-19 infection rose by 5% among pregnant women [119]. According to the findings of another piece of research (Zambrano et al., 2020), pregnant women who have COVID-19 have a threefold increased risk of being hospitalized in an intensive care unit (ICU) compared to women who are typically pregnant and need ample ventilation with oxygen. Several studies [120,121] report an increase in the number of instances of stillbirth and miscarriages that occurred during the pandemic. This was attributed to a decline in the quality of health treatment. The most likely cause is the physiological changes that occur during pregnancy as a result of the growth of the baby. These changes include a reduction in lung capacity, an increased risk of thrombosis, and a weakened immune system. These investigations show that the fetus only seldom is born with COVID-19. This may be because the virus does not induce significant levels of viremia and, as a result, does not pass through the transplacental membrane. In addition, another study [122,123,124] suggested that the absence of placental ACE2 and TMPRSS2 was to blame for this phenomenon. Breastfed infants are at an increased risk of contracting the disease because of their close relationship with an infected mother, even if they seem healthy and have no symptoms [125].

## 10. Impact of COVID-19 Vaccination on Pregnancy and Lactation

Even though there were little data available about the safety of any of the candidate vaccines on pregnancy or lactation for young babies, the Centers for Disease Control and Prevention (CDC), The Society for Maternal–Fetal Medicine (SMFM), and the American College of Obstetricians and Gynecologists (ACOG), all encouraged pregnant and nursing women to get the vaccine [126]. Following the COVID-19 RNA vaccine delivery to 150,000 pregnant women, there were no reports of any significant adverse reactions. However, some of them experienced significantly greater levels of injection site discomfort, fever, chills, headache, and muscular exhaustion compared to females who were not pregnant. The adverse effect observed after vaccination was the same one reported after receiving any other vaccine. In addition, maternal immunoglobulins G antibodies can efficiently cross the placenta, and as a result, it is theorized that COVID-19 vaccinations should be passed from mother to child during pregnancy. Regrettably, it is not known if the quantity of these antibodies is sufficient for adequate innate immunization [127,128].

Because milk is a barrier against the spread of infectious illnesses, breastfeeding is a crucial practice, particularly in the newborn’s first few days after birth. It has been estimated that breastfeeding may save 135 unexpected deaths in infants after delivery and lower the risk of sudden death in infants by 35%. Researchers have found evidence of COVID-19 antibodies, namely IgA and IgG, in the breast milk of mothers who have just recently recovered from the virus [129]. It is recommended that more studies be conducted to determine whether or if these newborns are immune to COVID-19 due to the antibodies they have been exposed to [130]. Two or three separate investigations have established the presence of viral RNA in the milk of infected mothers; however, the analysis did not focus on determining whether suckling infants acquired the infection via their mothers’ milk or from intimate contact with their infected mothers [131]. Since immunoglobulins may be transferred from a mother to her child via breast milk, several researchers have recommended that nursing mothers get the vaccination [127]. Some studies have shown a strong secretion of immunoglobulin A, immunoglobulin G, and immunoglobulin M in the milk of mothers six weeks following the injection of the vaccine.

## 11. Effect of COVID-19 Infection and Vaccine on Male Sex Hormones and Sexual Health

According to Lo et al.’s research from 2022, there is no evidence that SARS-CoV-2 was present in the sperm of infected or recovered males [132]. Despite this, contradictory results on the amount of testosterone were discovered. Several investigations have shown that testosterone levels stay steady. At the same time, other researchers have observed lower levels and linked this fall to the feverish condition of an infected male and said that it was transient [133,134]. Several types of research have shown erectile dysfunction (ED), and some of these studies have hypothesized that ED is caused by cardiac damage and myocarditis, both of which may result in a loss in blood flow and vascular function in the penis cavernous tissue [135]. Infected males have a series of physiological and biochemical changes, all of which may have a negative impact on libido. These include reduced function of the neurological system, respiratory issues, and altered hormone levels. In addition, the physiological state of the lockdown and the global economic situation generally affected the sexual desire of both people in couples, and some researchers examined the causes of the decline of sexual intercourse [136].

There was widespread concern about the impact that COVID-19 immunization might have on the quality of sperm and the ability to fertilize eggs, particularly from RNA vaccinations. A misconception led to the belief that the spike protein would be incorporated into the human genome and change gametes’ DNA. However, there is no mention of any occurrences of miscarriage or stillbirth after vaccination, nor any indication of post-vaccination sexual ineptitude or ED. After immunization, some of the candidates for the vaccine had renal irritation in very uncommon occurrences, but none of them reported changes in their sexual function [137,138].

Based on the information above, it is perfectly safe to take the COVID-19 vaccination. On the other hand, infection with COVID-19 may cause both temporary and permanent changes in the sexual function of both male and female hosts, and these changes can be caused either directly by the virus or indirectly by a malfunction in other bodily systems.

## 12. Conclusions

It has been demonstrated that females and males react inversely to numerous viral infections in various mammalian species. Gender influences both the incidence and severity of viral infections. It has been reported that males are more susceptible to this condition than females. This distinction results from a highly complex system incorporating behavioral, genetic, hormonal, and immunological components. According to this theory, the most recent data on COVID-19 mortality indicates that men are more susceptible to the disease than women. Complications, such as hypertension, diabetes, renal disease, and cardiovascular disease, are also strongly linked with the COVID-19 death rate. These disorders are closely related to smoking and alcohol consumption, significantly associated with males, particularly in African and Asian countries. Sex hormones and specific X-linked genes may also contribute to this difference in COVID-19 outcomes based on gender. These genes influence both innate and adaptive immune responses against viral infections. In addition, the androgen-independent expression of TMPRSS2 may contribute to the high prevalence of COVID-19 in males. Epidemiological data on SARS indicated that males were more susceptible to the disease. Thus, research on SARS-infected animals revealed that male rodents are more susceptible to infection with SARS-CoV-2 and confirmed the protective effect of estrogen signaling against the pathogenesis of SARS. Infection by COVID-19 showed several defects in sexual desire and sexual activity, such as ED. Infection increased the cases of stillborn and abortions and the risk of ICU admission for pregnant females. On the other hand, none of the candidate vaccines showed any effect on sexual activity for both genders. Figure 3 summarizes the impact of COVID-19 infection on males and females.

## Figures and Tables

**Figure 1 biomedicines-11-01677-f001:**
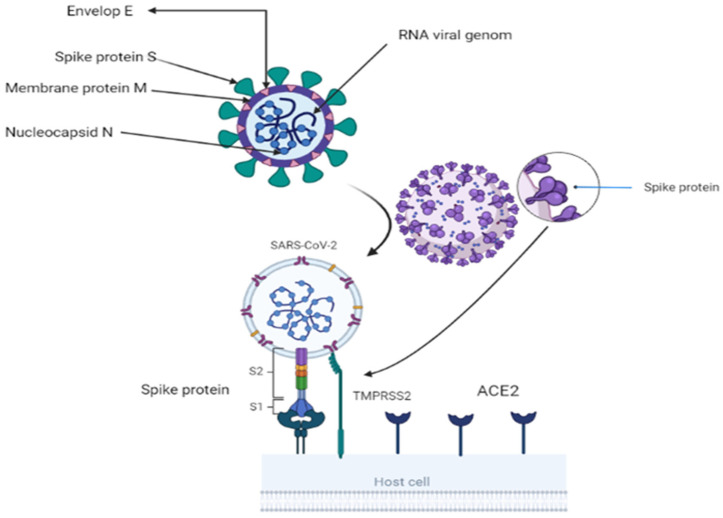
Showing structure of coronavirus and virus attachment using S2 fusion with cell receptor ACE2 and activation of S protein by TMBRSS2 through proteolytic cleavage. BioRender (https://app.biorender.com/) program is utilized for diagram creation.

**Figure 2 biomedicines-11-01677-f002:**
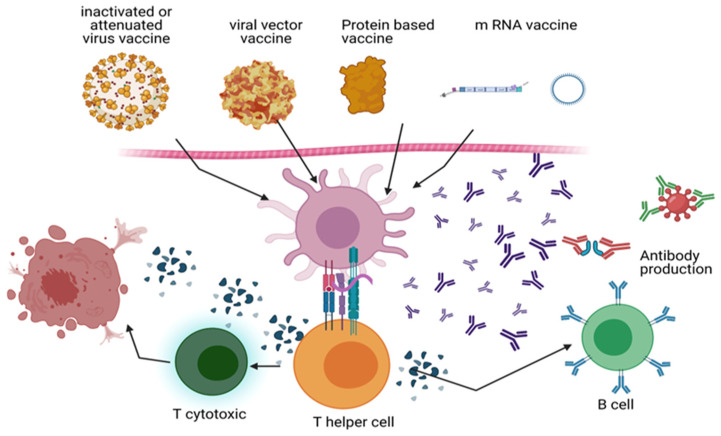
COVID-19 immunization strategies: The mechanism of action of inactivated vaccines (Sinopharm and Sinovac), viral vector vaccines (Oxford and AstraZeneca’s Johnson & Johnson Janssen’s Ad26.COV2.S), mRNA vaccines (Pfizer-BioNTech and Moderna), and recombinant DNA vaccines (Novartis and GSK-Sanofi) is depicted. During vaccination, dendritic cells and macrophages eat the virus or proteins interpreted by the viral genome. SARS-CoV-2 polypeptide produced on APCs stimulates T helper (Th) cells, which in turn activate B cells and cytotoxic T cells (Tc). By targeting the S protein, B cells generate antibodies that destroy viruses and other viral proteins. By cytolysis, Tc cells eliminate infected host cells. Memory B and T cells strengthen the immune system. BioRender generates diagrams (https://app.biorender.com/).

**Figure 3 biomedicines-11-01677-f003:**
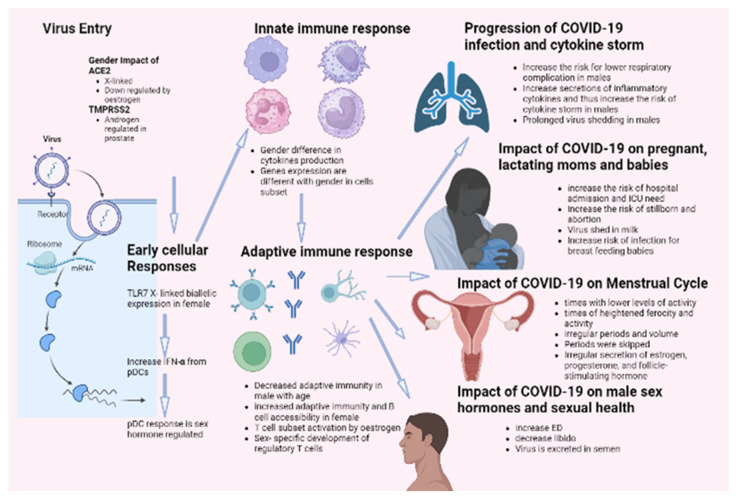
Impact of COVID-19 infection on gender difference.

## Data Availability

Not applicable.

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
