# Peer review of "Gender Differences in Response to COVID-19 Infection and Vaccination"

_biomedicines, 2023, doi:10.3390/biomedicines11061677_

Round 1
Reviewer 1 Report
The article is intriguing because it explores the gender differences in front of a COVID19 infection. It is interesting to note that ACE2 are expressed in pancreas too and a recent sudy has evidenced an increased pancreatic damages following SARS2 infections (see: Clin Gastroenterol Hepatol 2020, 18: 2128-2130) In that way, more males show a great risk of developing diabete or pancreatic cancer than female patients
Author Response
Thank you for your valuable recommendation. Also, we would like to thank you for your precious time for reviewing our manuscript.
We add this information Lines 271- 275 with reference no 60 (The correction is made in gray color).
Thank you
Reviewer 2 Report
The manuscript titled "Gender Differences in Response to Covid-19 Infection and Vaccination" deals with an interesting topic. The authors aim to synthesize current evidence on the role of COVID-19 infection and vaccination on male and female sexual activity, hormones, and the menstrual cycle for females, as well as on male sex hormones and sexual activity during infection and after vaccination.
The paper is well-written but it needs some changes and clarifications:
- The objective of the paper should be inserted in the "Introduction" section.
- In the section "Origen and Structure of SARS-CoV-2" must be explained the meanings of the abbreviations ORF, RBD, ERCIC, ER, GI, NTD, LKR.
- Line 94: please, specify the meaning of the abbreviation EUA.
- Line 282: please, remove the square bracket after the reference number 82… [82].].
- Line 284: please, specify the meaning of the abbreviation CCR5.
- Line 343: it is reported that "Changes in menstruation volume were found in 45 of the 177 people investigated for this study (25 percent)" but it is unclear for the reader which study it is. Please, explain better.
- Line 390: the sentence "Saudi research In Saudi Arabia, 4,170 people were asked for their feedback" is unclear. Please, write better.
Author Response
Thank you for your valuable recommendation. Also, we would like to thank you for your precious time reviewing our manuscript. All corrections are made in yellow color
- The objective of the paper should be inserted in the "Introduction" section.
Done line 43-46
- In the section "Origen and Structure of SARS-CoV-2" must be explained the meanings of the abbreviations ORF, RBD, ERCIC, ER, GI, NTD, LKR.
All Done
ORF: line 55
NTD: was already there lines 65-66
RBD: line 67
ERGIC: line 71
ER: line 72
GI: line 72
LKR: line 78
- Line 94: please, specify the meaning of the abbreviation EUA.
Done lines: 100-101
- Line 282: please, remove the square bracket after the reference number 82… [82].].
Done Line 476 (now after modification)
- Line 284: please, specify the meaning of the abbreviation CCR5.
Done it became line 478 after modification.
- Line 343: it is reported that "Changes in menstruation volume were found in 45 of the 177 people investigated for this study (25 percent)" but it is unclear for the reader which study it is. Please, explain better.
In the study cited by [95], changes in menstruation volume were found in 45 of the 177 people investigated for this study (25 percent). 36 out of these 45 (25%) had a noticeably heavier period. Done by adding, “mentioned” line 539
- Line 390: the sentence "Saudi research In Saudi Arabia, 4,170 people were asked for their feedback" is unclear. Please, write better.
Done line 585
Thank you
Reviewer 3 Report
In the review named “Gender Differences in Response to Covid-19 Infection and Vac-2 cination” author analyzed the role of COVID-19 infection and vaccination on male and female sexual activity, hormones, and the menstrual cycle for females, as well as on male sex hormones and sexual activity during infection and after vaccination. Although this paper give some important information and make a good revision some minor questions are need to solved
1)In figure 1 The fusion between viral particle and ACE2 receptor seems to be out of focus and hard to see.
2)In Vaccination section, authors do not talk about Astra-ZENECA vaccine however in countries as Spain this vaccine was accepted and peoples received two doses.
3)Author only talk about vaccines approved in US and other countries as China but it is important give a general vision around the world about the vaccination state.
4)In figure 2 perhaps DNA vaccine must be changes by protein based vaccines because none of them is made with DNA.
5)In point 3.2 again Astra-Zeneca vaccine is not included. And the some vaccine adverse effects are missing.
6)In point 3.3 information about GSK vaccine is missing.
7)In point 4 a separation between effects in each gender will be necessary to better understand this point
8)Perhaps some points where the effect of COVID-19 infection and vaccination on teenagers will be appropriated. This is because in these ages the female and male hormones are altered.
9)In both points 5 and 6 none are discussed about especial hormone situations as prostate cancer or breast cancer, hormone therapy, quimiotherapy...
10)In line 206 a reference is missing
11)Figure 13 is again out of focus and hard to see the letter
Minor editing of English language required
Author Response
Thank you for your valuable recommendation. Also, thank you for your precious time reviewing our manuscript. All corrections are made in turquoise color
1)In figure 1 The fusion between viral particle and ACE2 receptor seems to be out of focus and hard to see.
Modifications has been performed to clarify the fusion between viral particle and ACE2 receptor
2)In Vaccination section, authors do not talk about Astra-ZENECA vaccine however in countries as Spain this vaccine was accepted and peoples received two doses.
Done Lines 115- 147
3)Author only talk about vaccines approved in US and other countries as China but it is important give a general vision around the world about the vaccination state.
Done during the explanation of Astra Zeneca status lines 113-115
4)In figure 2 perhaps DNA vaccine must be changes by protein-based vaccines because none of them is made with DNA
We strongly agree but that how researches have classified the vaccine platform and according to the manufacturer descriptions and approval. But We have made the modification as we strongly agree.
5)In point 3.2 again Astra-Zeneca vaccine is not included. And some vaccine adverse effects are missing.
It is included line 162. The adverse effect of Astra-Zeneca is now included lines: 115- 147. For inactivated vaccines lines: 199-206.
6)In point 3.3 information about GSK vaccine is missing.
Done : lines 182- 186
7)In point 4 a separation between effects in each gender will be necessary to better understand this point
Done section 4.1 and 4.2 is added
8)Perhaps some points where the effect of COVID-19 infection and vaccination on teenagers will be appropriated. This is because in these ages the female and male hormones are altered.
Done section 4.3
9)In both points 5 and 6 none are discussed about especial hormone situations as prostate cancer or breast cancer, hormone therapy, chemotherapy...
For prostate cancer and hormones lines 276-287
Breast cancer lines 320-323
hormonal therapy lines: 323-328
Chemotherapy lines: 329- 341
10)In line 206 a reference is missing
Done: line 260
11)Figure 13 is again out of focus and hard to see the letter.
We tried to enlarge the figure so that the letter could be seen
Thank you